# Effect of habitual reading direction on saccadic eye movements: A pilot study

**Anqi Lyu**[1,2], **Larry Abel**[2,3], **Allen M. Y. Cheong**[1,4]*

**1** School of Optometry, The Hong Kong Polytechnic University, Kowloon, Hong Kong, **2** Department of Optometry and Vision Sciences, The University of Melbourne, Melbourne, Victoria, Australia, **3** School of Medicine, Deakin University, Melbourne, Victoria, Australia, **4** Research Centre for SHARP Vision (RCSV), The Hong Kong Polytechnic University, Kowloon, Hong Kong

* allen.my.cheong@polyu.edu.hk

**Data Availability Statement:** The data supporting the finding of this study are openly available at: https://figshare.com/articles/dataset/Effect_of_habitual_reading_direction_on_saccadic_eye_movements_A_pilot_study/19841758.

## Abstract

Cognitive processes can influence the characteristics of saccadic eye movements. Reading habits, including habitual reading direction, also affect cognitive and visuospatial processes, favouring attention to the side where reading begins. Few studies have investigated the effect of habitual reading direction on saccade directionality of low-cognitive-demand stimuli (such as dots). The current study examined horizontal prosaccade, antisaccade, and self-paced saccade in subjects with two primary habitual reading directions. We hypothesised that saccades responding to the stimuli in subject's habitual reading direction would show a longer prosaccade latency and lower antisaccade error rate (errors being a reflexive glance to a sudden-appearing target, rather than a saccade away from it). Sixteen young Chinese participants with primary habitual reading direction from left to right and sixteen young Arabic and Persian participants with primary habitual reading direction from right to left were recruited. All subjects spoke/read English as their second language. Subjects needed to look towards a 5˚/10˚ target in the prosaccade task or look towards the mirror image location of the target in the antisaccade task and look between two 10˚ targets in the self-paced saccade task. Only Arabic and Persian participants showed a shorter and directional prosaccade latency towards 5˚ stimuli against their habitual reading direction. No significant effect of reading direction on antisaccade latency towards the correct directions was found. Chinese readers were found to generate significantly shorter prosaccade latencies and higher antisaccade directional errors compared with Arabic and Persian readers for stimuli appearing at their habitual reading side. The present pilot study provides insights into the effect of reading habits on saccadic eye movements of low-cognitive-demand stimuli and offers a platform for future studies to investigate the relationship between reading habits and eye movement behaviours.

## 1. Introduction

### 1.1 Saccadic eye movements

Humans do not look at a scene with steady gaze. Our eyes move around, bringing the interesting parts of the scene to the fovea with a frequency of 2 or 3 fixations per second [1]. In fact, saccadic eye movement is one of the fastest movements produced by the human body, serving

**Funding:** This work was supported by Hong Kong SAR Government General Research Fund (15102717) and The Hong Kong Polytechnic University Departmental Research Grant (G-UAG2) to Cheong AMY, and Departmental Postgraduate Research Studentship to Lyu A. The funders had no role in study design, data collection and analysis, decision to publish, or preparation of the manuscript.

**Competing interests:** The authors have declared that no competing interests exist.

in bringing the images of objects of interest into the central vision for detailed analysis. The perception of the environment relies on saccades and fixations which are the stops in-between saccades [2]. A distributed network including cortical (mainly frontal and parietal) and sub-cortical (basal ganglion, superior colliculus, midbrain, brain stem, thalamus and cerebellum) areas are involved in generating saccades [3]. It has been suggested that understanding the saccadic system provides researchers with a valuable "microcosm of the brain" as its input can be controlled and manipulated, while its output can be accurately recorded and quantified using different experimental paradigms [4]. A range of eye movement tasks have been used in the literature to examine the characteristics of saccades, including prosaccade, antisaccade and self-paced saccade tasks. Prosaccades, which are also known as reflexive saccades, examine the response time (latency) and accuracy of a saccade (saccade gain in terms of the ratio of saccade amplitude / target amplitude) to a sudden-onset peripheral visual stimulus. An antisaccade requires the suppression of a reflexive saccade towards a sudden-onset stimulus and the execution of a voluntary saccade to the opposite direction of the stimulus. The parallel nature of antisaccade programming assumes a competition arises between the exogenously triggered prosaccade and the endogenously initiated antisaccade at the onset of stimulus [5–7]. For example, if the exogenously triggered prosaccade is programmed too fast (or the endogenously initiated antisaccade is too slow to reach the threshold for activation), it "wins" the competition and makes a reflexive saccade first (i.e. antisaccade directional error), followed by a corrective antisaccade [8]. Directional error rate (i.e. the proportion of glances towards the stimulus) and the latency of correct responses are commonly analysed in the antisaccade task. Self-paced saccade task has been considered as an almost entirely volitional eye movement task that requires repetitive and self-initiated refixations between two static visual stimuli [9]. A vast array of studies have suggested that many cognitive processes, including those involved in attention [10–12], working memory [13] and learning [14], have an impact on the characteristics of saccadic eye movements.

## 1.2 Effect of cognitive process on saccadic eye movement

Attention is needed to orient the target location prior to the execution of a saccade [12]. Saslow reported a decrease in prosaccade latency from 200 msec to 150 msec if a stimulus appeared 200 msec or longer after the termination of the fixation, compared with the situation where the offset of fixation and onset of stimulation occurred simultaneously [15]. By introducing a medium temporal gap (200–250 msec) between the offset of a central fixation target and the onset of a peripheral stimulus, Fischer and Weber found a significant decrease in antisaccade latency but a significant increase in antisaccade error rate [10]. One explanation for these saccade changes was that this temporal gap contributed to the disengagement of attention before the target appeared. Moreover, studies manipulating the likelihood of a target presenting on either left or right side of a central fixation point found that subjects showed shorter prosaccade latency in responding to the target with a higher probability of presentation. This suggested an effect of learning in modifying the prosaccade performance [14, 16].

While changing the direction of letters and words within English sentences (i.e. both letters and words were orientated from right to left), Inhoff et al. reported less efficient saccadic eye movements in English readers compared with their reading normal English texts. However, these performances improved with practice [17]. In addition to these studies, extensive findings have revealed a wide range of cognitive processes influencing saccadic eye movements [10–12]. Even a simple prosaccade involves a complex weighting of both bottom-up information (stimulus properties) and top-down information (cognitive factors), although the precise nature for the degrees of the control remains unclear [8]. Nevertheless, our cognitive systems

are shaped / influenced by cultural practices such as reading habits (see section 1.3 below), which suggests a potential impact of reading habits on characteristics of saccadic eye movements.

## 1.3 Effect of habitual reading direction on cognitive systems

Han and Northoff (2008a) provided neuroimaging evidence that transcultural differences could affect the neural activities underlying both high-level and low-level cognitive functions [18]. They proposed to investigate the influence of reading direction on regulating the functional organization of the human brain as well as related neurocognitive processes [19]. Reading direction has been found to influence many cognitive functions, such as directional differences in facial expression perception [20], aesthetic preference [21] and utilization of visual space [22]. Especially, visuospatial attention could be modulated by the habitual reading direction. During a letter matching task, English readers with the habitual reading direction of left-to-right (LTR) were found to spend a longer time in responding to the stimuli in the right visual field, while Hebrew readers with the habitual reading direction of right-to-left (RTL) took longer to respond to the stimuli that appeared in the left visual field. It was suggested that reflexive attention showed biases on the side where reading began [23]. Consistent with this study, several studies have confirmed the impact of habitual reading direction on the asymmetries of visuospatial attention [24–26]. For example, Rinaldi et al. (2014) compared the performance on a star cancellation task between Italian and Israeli subjects who were instructed to mark the small stars amongst many randomly distributed distractors (large stars, English or Hebrew letters and words). They found that monolingual Italian subjects (i.e. reading from LTR) made more omissions in the right visual field, while monolingual Israeli subjects (i.e. reading from RTL) omitted more targets in the left visual field [25]. However, bilingual subjects who managed reading in both directions did not show any spatial asymmetries. Furthermore, Afsari et al. (2016) examined the effect of habitual reading direction on visual exploration in a group of bilingual readers of a native LTR language and a secondary RTL language. They found that native LTR readers who studied a secondary RTL language in late life showed a leftward bias with more fixations on the left part of a natural image, and this horizontal bias in the exploration of images did not alter when they first read either LTR or RTL text primes [26].

In addition to the biased visuospatial attention, we questioned whether reading direction also contributed to the left-right asymmetry of saccadic eye movements. While reading continuous text, LTR texts such as English [27] and German [28] elicit saccades towards the location slightly to the left of a word centre, while RTL scripts such as Hebrew [29] and Uighur [30] have saccades landing to the position slightly to the right of a word centre. In addition to saccades generated towards the left and right directions, Yan et al. took the advantage of the fact that Chinese text can be oriented horizontally and vertically without disturbing the shape of characters and reported a similar saccadic landing position for 28 young readers reading horizontal and vertical Chinese texts [31]. These studies demonstrated that reading direction affected saccadic eye movements during high-level reading processes. Nevertheless, fewer studies have investigated the impact of the habitual reading direction on the directionality of saccadic eye movements during low-cognitively demanding tasks such as responding to a dot. Most of the studies investigating the left-right asymmetry of these saccadic eye movements focused on ocular dominance [32–34] or hand dominance [35, 36]. Understanding the effect of habitual reading direction on saccadic eye movements to low-cognitive-demand stimuli would help researchers to better study the impact of reading direction on saccade generation

between populations, and to provide insights into cultural influences on the modulation of saccadic eye movements and spatial attention.

In the present pilot study, young bilingual healthy participants with two primary habitual reading directions were recruited to complete 3 types of saccadic eye movement tasks, namely horizontal prosaccade, antisaccade and self-paced saccade. These encompassed both reflexively and volitionally initiated saccades. We hypothesized that readers who habitually read from LTR would show a leftward asymmetry in the saccadic parameters (i.e. shorter prosaccade latency and higher antisaccade error rate) when they made a saccade to a target appearing on the side where reading began (i.e. the side where reflexive attentional orientation was biased, in this case on the left of a fixation point). When the target appeared on the side of their habitual reading direction (i.e. to the right of the fixation point), we expected longer prosaccade latencies and lower antisaccade error rates. In contrast, readers who habitually read from RTL would show a rightward asymmetry.

## 2. Material and methods

### 2.1 Subjects

32 young university students who were bilingual speakers / readers were recruited (20 were recruited and tested at The University of Melbourne and 16 were recruited and tested at The Hong Kong Polytechnic University). 16 subjects were Chinese readers whose primary reading direction was LTR and 16 were Arabic and Persian readers (12 Arabic and 4 Persian) whose primary reading direction was RTL. All participants were aged between 18 and 35 years old with normal or corrected to normal vision and started to learn English (with reading direction of LTR) since their early childhoods. Subjects were separated into 2 groups according to their primary habitual reading direction. The characteristics of the participants are shown in Table 1. To control the potential confounding influence of the educational level, this factor was controlled and matched between these 2 groups. Participants in the RTL group were significantly older than the LTR group (Student's t-test: 23.81 ± 3.43 vs. 27.13 ± 3.83 years, t(30) = 2.58, p = 0.02). Nevertheless, horizontal saccade latency is relatively stable from age of 14 to 50 years old [37], so this factor should not be a concern. Exclusion criteria were any history of ophthalmic, neurological or psychotic illness, or any medication intake that might affect eye movements. The experiment was approved by The University of Melbourne human research ethics committee (HREC #1647981.1) and Department of Research Committee of the School of Optometry of The Hong Kong Polytechnic University (HSEARS20191217001). All participants gave written informed consent. The study followed the tenets of the Declaration of Helsinki.

### 2.2 Apparatus and stimuli

As the data were collected at 2 sites, minor differences in the experimental setting were present, including display monitor and testing distance, whereas the stimulus size and distance

**Table 1. Descriptive characteristics of participants.**

| | Chinese participants (N = 16) | Arabic (N = 12) and Persian (N = 4) participants | P-value |
|---|---|---|---|
| Habitual reading direction | From left to right (LTR) | From right to left (RTL) | |
| Age (y) | | | 0.02 |
| mean (SD) | 23.81 (3.43) | 27.13 (3.83) | |
| range (y) | 19–32 | 18–34 | |
| English education (y), mean (SD) | 16.81 (3.27) | 16.56 (6.41) | 0.89 |
| range (y) | 10–21 | 6–30 | |

were adjusted so that the same visual angle was elicited. The testing sequence and programs used (including the monitors' resolution (1920 x 1080) and refresh rate (60 Hz)) were identical between the 2 sites. The stimulus was a 1° black dot against a white background on a 27-inch (U2711B, Dell Technologies, Round Rock, Texas, United States) or a 24-inch LCD monitor (BENQ xl2540) in Melbourne and Hong Kong sites respectively. Participants sat comfortably at 75 cm (Melbourne) or 65 cm (Hong Kong) in front of the monitor with their chin resting on a chinrest to stabilize their head position. Movements of both eyes were recorded using an infrared video eye tracking system (Eyelink 1000 or Eyelink Portable Duo, SR Research, Scarborough, ONT, Canada) with a sampling rate of 500 Hz, in which there was no statistical difference between the results from the two eye trackers (see S1 Appendix). Subjects were asked to perform the following eye movement tasks.

## 2.3 Procedures

Participants' eye movements were assessed while conducting 3 visual tasks: 1) prosaccade, 2) antisaccade, and 3) self-paced saccade tasks. The tasks were performed sequentially in 3 blocks.

Stimuli were presented pseudorandomly in locations 5° or 10° to the left or right of the centre of the monitor. Participants were instructed to fixate at a central cross and then to look towards the stimulus in the prosaccade task (see Fig 1A) or look towards the mirror image location of the stimulus in the antisaccade task (see Fig 1B) as soon as the stimulus was presented and fixation disappeared. Fifty-two trials were conducted in the prosaccade task to assess the prosaccade latency (i.e. reaction time responding to the onset of stimuli) and gain (i.e. ratio of saccadic amplitude to stimuli amplitude). Express saccades whose latency fell between 80 to 120 msec [38–40] were excluded from the analysis. Less than 10% of the trials were excluded for all participants. Fifty-two trials were conducted in the antisaccade task to assess the antisaccade latency of correct responses (i.e. saccades made to the correct direction) and error rate (i.e. proportion of prosaccade errors). In the self-paced saccade task, two stimuli were shown for 45 seconds at 10° left and right of the monitor centre. Participants needed to look back and forth between these two dots as rapidly and as accurately as possible for the entire duration of the task. Gain (i.e. ratio between the primary saccades and target amplitudes) and inter-saccadic intervals (ISI) (i.e. interval between onset of the saccades) were collected and submitted to data analysis.

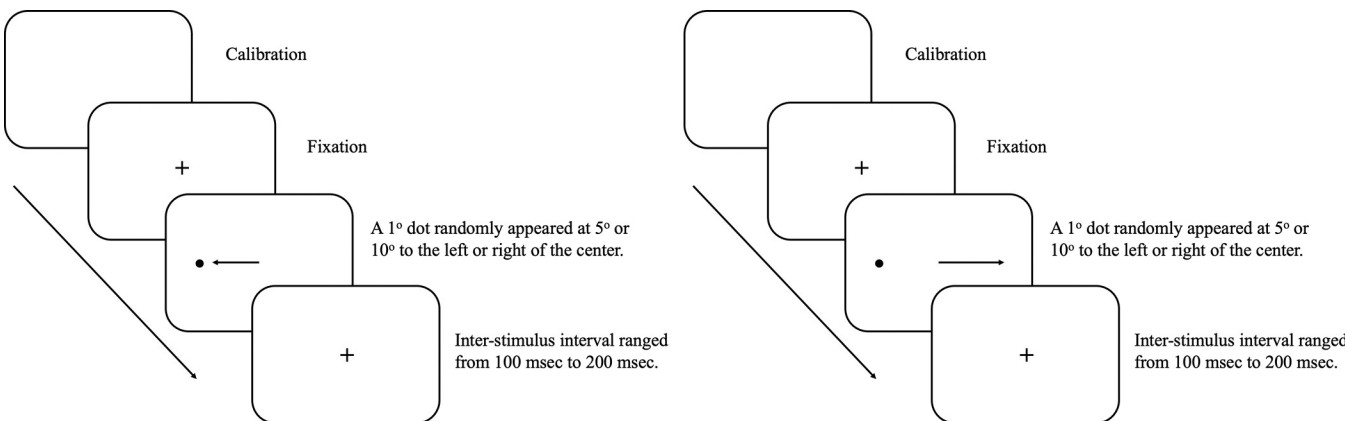

**Fig 1. Sample trial of prosaccade and antisaccade task.** (a) A sample trial of a prosaccade task. Participants needed to make a saccade towards the stimulus as quickly as possible. (b) A sample trial of an antisaccade task. Participants needed to look at the mirror image of the stimulus location.

## 2.4 Data analysis

All statistical analyses were performed using GraphPad Prism version 9.2.0.332 for Windows (GraphPad Software, San Diego, California USA, www.graphpad.com). Eye movement parameters were not significantly different from normal distribution (Kolmogorov-Smirnov goodness of fit test, p>0.05). Dependent variables (prosaccade latency, prosaccade gain, correct antisaccade latency, antisaccade error rate, inter-saccadic interval, and gain in self-paced saccades) were analysed using analysis of variance (ANOVA) with group (LTR (Chinese participants) vs. RTL (Arabic and Persian participants)) as between-subject factor and the direction of stimuli (with- vs. against-habitual reading direction) and/or the magnitude of stimuli (5° vs. 10° targets) as within-subject factors, to assess any significant effect or interaction. In addition, the interaction effect between groups and saccade types (prosaccade vs. antisaccade) on saccade latency was examined using a 2-way ANOVA with group as a between-subject factor and saccade type as a within-subject factor. Any significance was examined using Bonferroni multiple comparison post hoc tests. Adjusted p-value of less than 0.05 was considered statistically significant.

## 3. Results

### 3.1 Effects of habitual reading direction on prosaccade eye movements

No significant main effect of group ($F(1, 30) = 3.27$, $p = 0.08$), stimulus magnitude ($F(1, 30) = 0.22$, $p = 0.64$) or stimulus direction ($F(1, 30) = 1.08$, $p = 0.31$) was found on prosaccade latency. The interactions between group and stimulus magnitude ($F(1, 30) = 1.00$, $p = 0.33$), and between stimulus direction and magnitude ($F(1, 30) = 0.51$, $p = 0.48$) were not significant. Nevertheless, there were significant interaction effects between group and stimulus direction ($F(1, 30) = 5.45$, $p = 0.03$) and among group, stimulus direction and magnitude ($F(1, 30) = 5.36$, $p = 0.03$) (Fig 2). LTR participants had shorter prosaccade latency compared with the RTL group when the stimuli appeared along their habitual reading direction (171.88 ± 20.21 vs. 194.66 ± 30.39 msec, $p = 0.001$), whereas no significant difference was found when the stimuli appeared against their habitual reading direction (176.96 ± 30.83 vs. 181.46 ± 21.44 msec, $p = 0.99$). When the stimuli were presented at 5° with-direction, the LTR group generated significantly shorter prosaccade latency compared with the RTL group (167.75 ± 22.06 vs. 198.98 ± 31.76 msec, $p = 0.01$). No group difference was found for 5° against-direction stimuli (176.56 ± 24.64 vs. 178.77 ± 23.86 msec, p>0.99). Furthermore, the RTL participants had a longer prosaccade latency when the 5° stimuli appeared along their habitual reading direction (i.e. stimuli at the left side of the fixation), compared with that which appeared against their reading direction ($p = 0.03$), whereas such difference was not found in the LRT group (p>0.99).

Prosaccade gain examines the accuracy of the landing positions of prosaccades. Statistically, group ($F(1, 30) = 0.48$, $p = 0.49$) and stimulus direction ($F(1, 30) = 0.42$, $p = 0.52$) had no effect on prosaccade gain, whereas a significant main effect of stimulus magnitude ($F(1, 30) = 4.75$, $p = 0.04$) was found. Similarly, there were significant interactions between the group and stimulus direction ($F(1, 30) = 4.26$, $p = 0.05$), and among group, stimulus magnitude and direction ($F(1, 30) = 5.52$, $p = 0.03$). No interaction effect between group and stimulus magnitude ($F(1, 30) = 1.75$, $p = 0.20$) or between stimulus magnitude and direction ($F(1, 30) = 0.14$, $p = 0.71$) was found (Fig 3). Participants in both groups generated more accurate prosaccade gain to 5° stimuli compared with 10° stimuli (1.00 ± 0.10 vs. 0.98 ±0.07 and 1.00 ± 0.18 vs. 0.95 ± 0.10 for the LTR and RTL groups respectively). The RTL group tended to have more accurate gain when the stimuli appeared against their habitual reading direction (i.e. stimuli appeared at the right side of fixation), compared with that which appeared along their reading direction, although this did not reach a significance level (0.99 ± 0.18 vs. 0.95 ± 0.09, $p = 0.07$). No

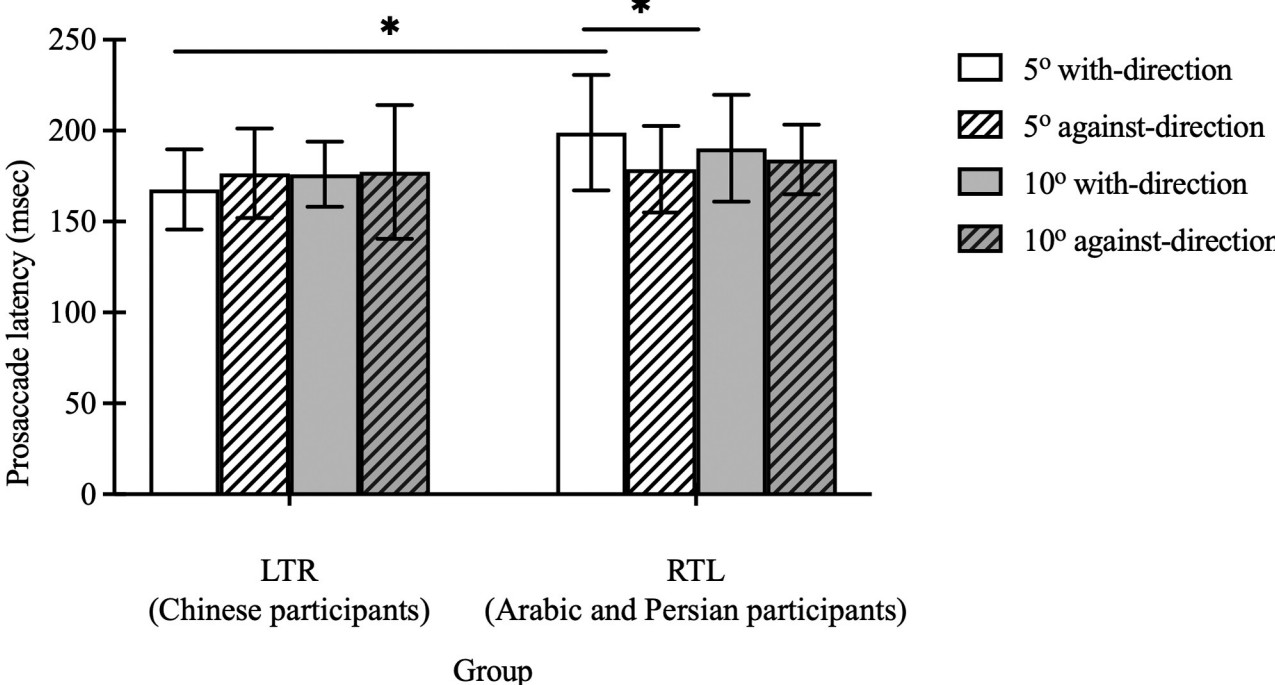

**Fig 2. Prosaccade latency towards 2 directions in LTR (Chinese participants) and RTL (Arabic and Persian participants) groups.** The prosaccade latency for the LTR and RTL groups when stimuli appeared at 5˚ with-direction, 5˚ against-direction, 10˚ with-direction, and 10˚ against-direction locations. Each bar represents the average prosaccade latency with standard deviation as the error bar. Black horizontal lines are the significant difference between different groups or the significant difference between 2 stimuli directions: *: p<0.05.

directional difference was found for the LTR group (1.01 ± 0.10 vs. 0.98 ± 0.07, p = 0.52). When the stimuli were presented against the subjects' reading direction, the RTL group generated a larger gain in responding to the 5˚ targets, compared with the 10˚ targets (1.03 ± 0.23 vs. 0.95 ± 0.11, p = 0.03), while no magnitude difference was found in the LTR group (0.98 ± 0.08 vs. 0.97 ± 0.05, p>0.99).

### 3.2 Effects of habitual reading direction on antisaccade eye movements

Opposite to the findings in prosaccade eye movements, there was no significant main effect or interaction effect of group, stimulus direction, or magnitude on antisaccade latency for the correct trials (F(1, 30)<0.79, p>0.38) (Fig 4).

When comparing antisaccade error rates between groups and among different stimuli conditions, a significant main effect of stimulus magnitude was found (F(1, 30) = 21.90, p<0.001), whereas group (F(1, 30) = 0.64, p = 0.43) and stimulus direction (F(1, 30) = 0.01, p = 0.94) did not significantly affect the rate of antisaccade directional errors. A significant interaction between group and stimulus direction was observed (F(1, 30) = 6.27, p = 0.02). No interaction between group and stimulus magnitude, stimulus magnitude and direction or among group, stimulus magnitude and direction was found (F(1, 30)<2.12, p>0.16). Both groups made more antisaccade errors towards 5˚ stimuli compared with 10˚ stimuli (0.24 ± 0.19 vs. 0.17 ± 0.17 and 0.24 ± 0.20 vs. 0.10 ± 0.12 for the LTR and RTL groups respectively). The LTR group made more directional errors for stimuli appearing at their habitual reading side compared with the RTL group (0.24 ± 0.18 vs. 0.14 ± 0.16, p = 0.05). No group difference was found when the stimuli appeared against their reading direction (0.17 ± 0.19 vs. 0.21 ± 0.19, p = 0.89).

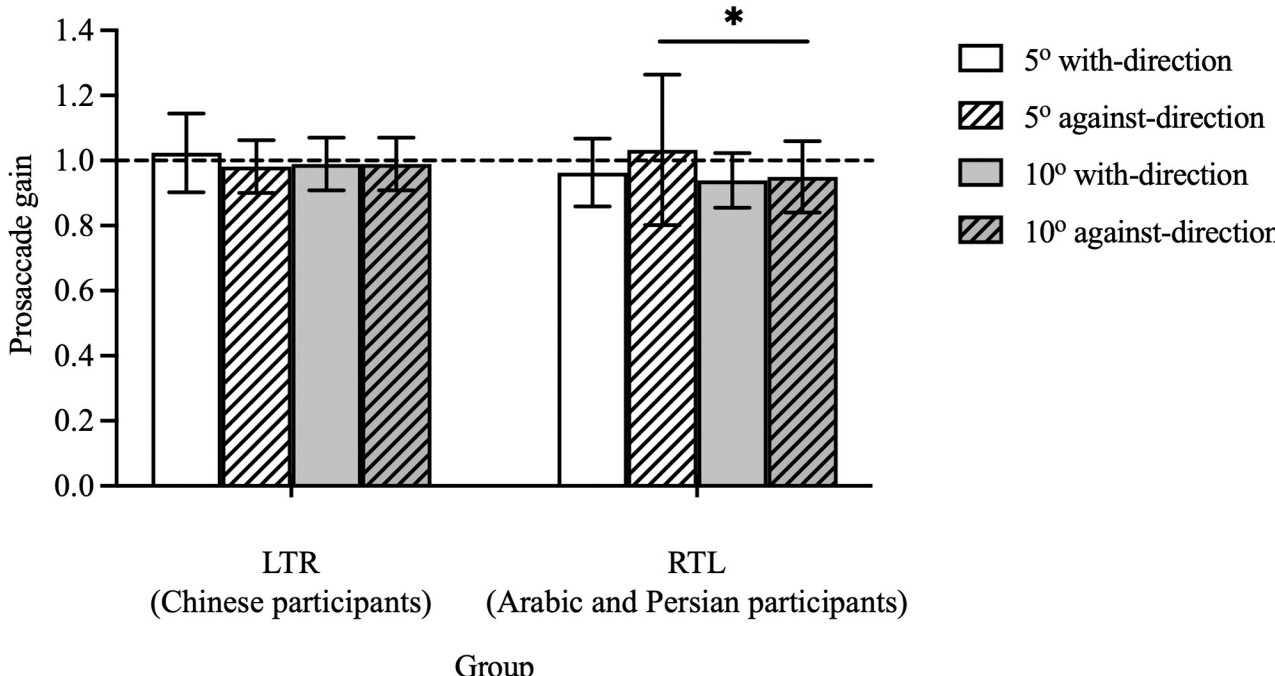

**Fig 3. Prosaccade gain towards 2 directions in LTR (Chinese participants) and RTL (Arabic and Persian participants) groups.** The prosaccade gain for the LTR and RTL groups when stimuli appeared at 5˚ with-direction, 5˚ against-direction, 10˚ with-direction, and 10˚ against-direction locations. Each bar represents the average prosaccade gain with standard deviation as the error bar. Black horizontal dotted line represents a gain equal to 1, where the prosaccade lands exactly on the target. Black horizontal line is the significant difference between stimuli magnitudes.

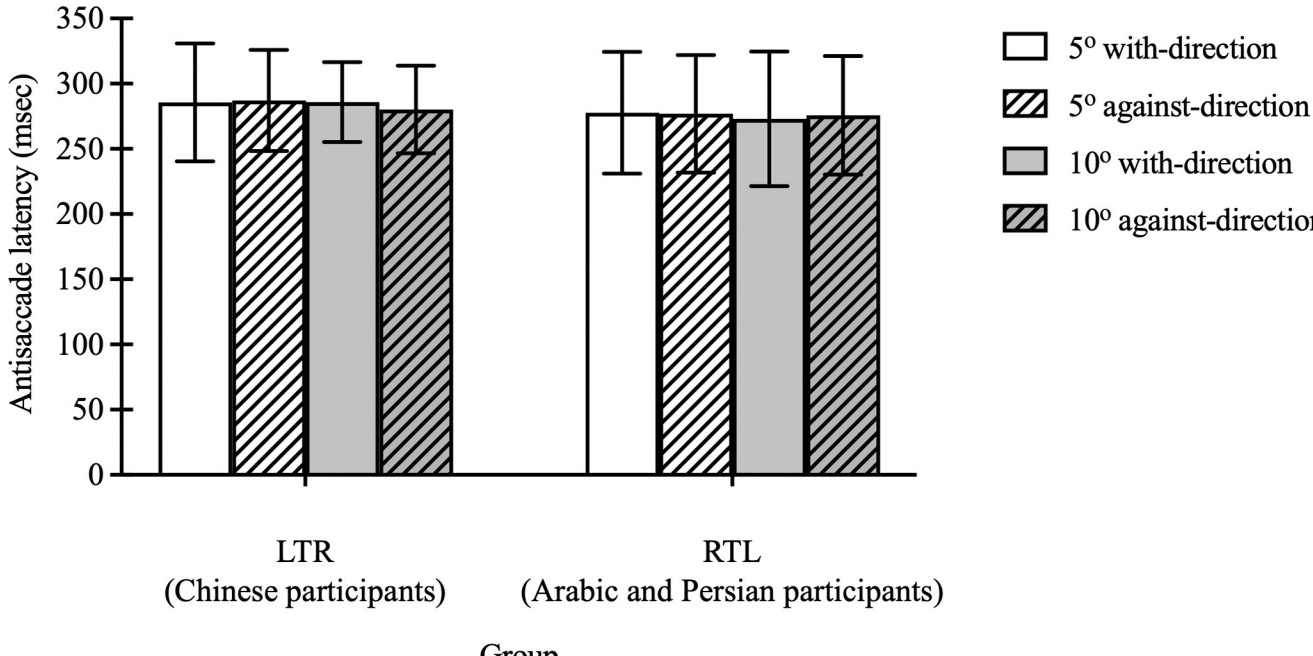

**Fig 4. Antisaccade latency of correct trials towards 2 directions in LTR (Chinese participants) and RTL (Arabic and Persian participants) groups.** The antisaccade latency for the LTR and RTL groups when stimuli appeared at 5˚ with-direction, 5˚ against-direction, 10˚ with-direction, and 10˚ against-direction locations. Each bar represents the average antisaccade latency with standard deviation as the error bar.

### 3.3 Effect of saccade type on saccade latency

Further analysis was performed to examine the effect of types of saccades (prosaccade vs. antisaccade) on saccade latency between groups. A significant effect of saccade type was found, with both groups' participants generating shorter prosaccades latency compared to antisaccades (F(1, 30) = 204.20, p<0.001; 174.42 ± 25.98 vs. 284.75 ± 36.69 and 188.06 ± 26.92 vs. 275.88 ± 46.18 msec for the LTR and RTL groups respectively) (Fig 5). No effect of groups (F(1, 30) = 0.08, p = 0.78) or interaction (F(1, 30) = 2.64, p = 0.11) was found.

### 3.4 Effects of habitual reading direction on self-paced saccade eye movements

Inter-saccadic interval (ISI) and gain were compared between groups and between the 2 directions of self-paced saccades (i.e. saccades towards habitual vs. non-habitual reading direction). There was no significant effect of group (F(1, 30)<0.01, p = 0.98), saccade direction (F(1, 30) = 0.14, p = 0.71) or interaction (F(1, 30) = 0.03, p = 0.86) on ISI. Both groups generated similar ISI towards the stimulus located at their habitual and non-habitual reading side (509.77 ± 172.37 vs. 514.57 ± 174.63 msec and 512.55 ± 120.98 vs. 514.29 ± 113.30 msec for the LTR and RTL groups respectively).

For self-paced saccade gain, no significant main effect of group (F(1, 30) = 0.20, p = 0.66) or saccade direction (F(1, 30) = 0.05, p = 0.83) was found, whereas a significant interaction between the 2 variables was observed (F(1, 30) = 14.19, p<0.001) (Fig 6). The Chinese participants showed more accurate gain when they made saccades to the stimulus located at the side of their habitual reading direction (i.e. the stimulus at the right side of the monitor) compared with the stimulus located in their non-habitual reading direction (1.01 ± 0.06 vs. 0.95 ± 0.03, p = 0.02), whereas Arabic and Persian participants generated more accurate saccade gain

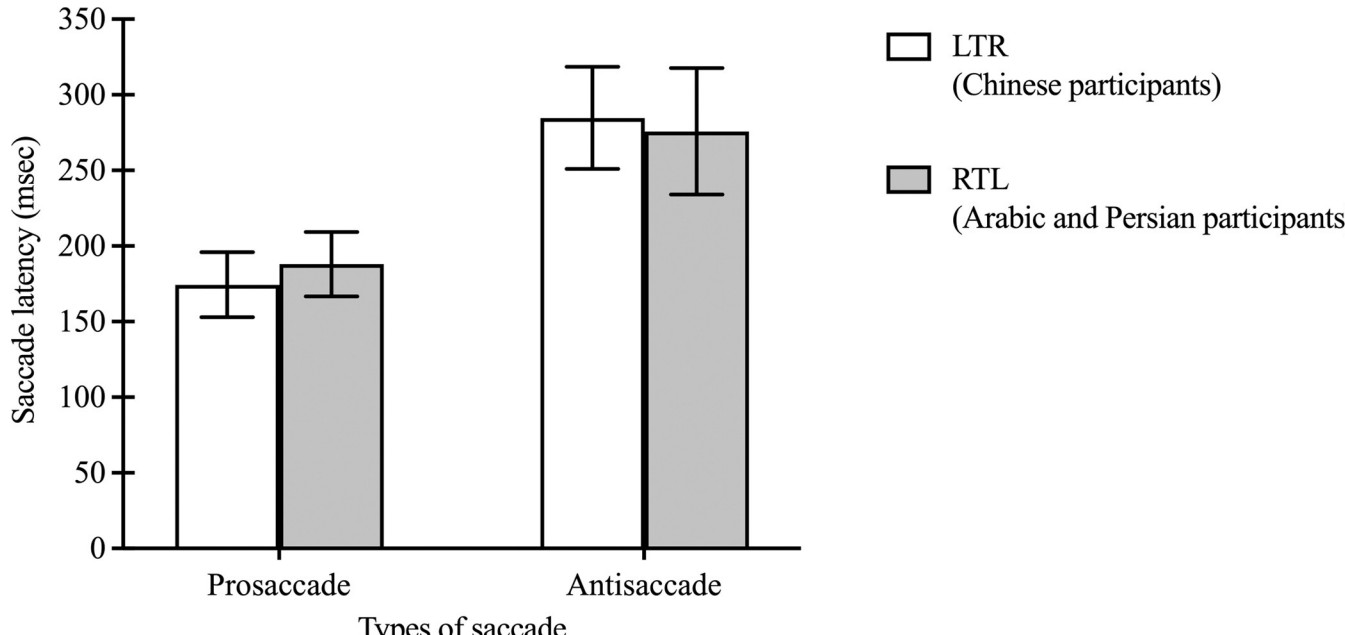

**Fig 5. Prosaccade latency and correct antisaccade latency of LTR (Chinese participants) and RTL (Arabic and Persian participants) groups.** The prosaccade and antisaccade latency for the Chinese as well as the Arabic and Persian groups. Each bar represents the average saccade latency with standard deviations as the error bar.

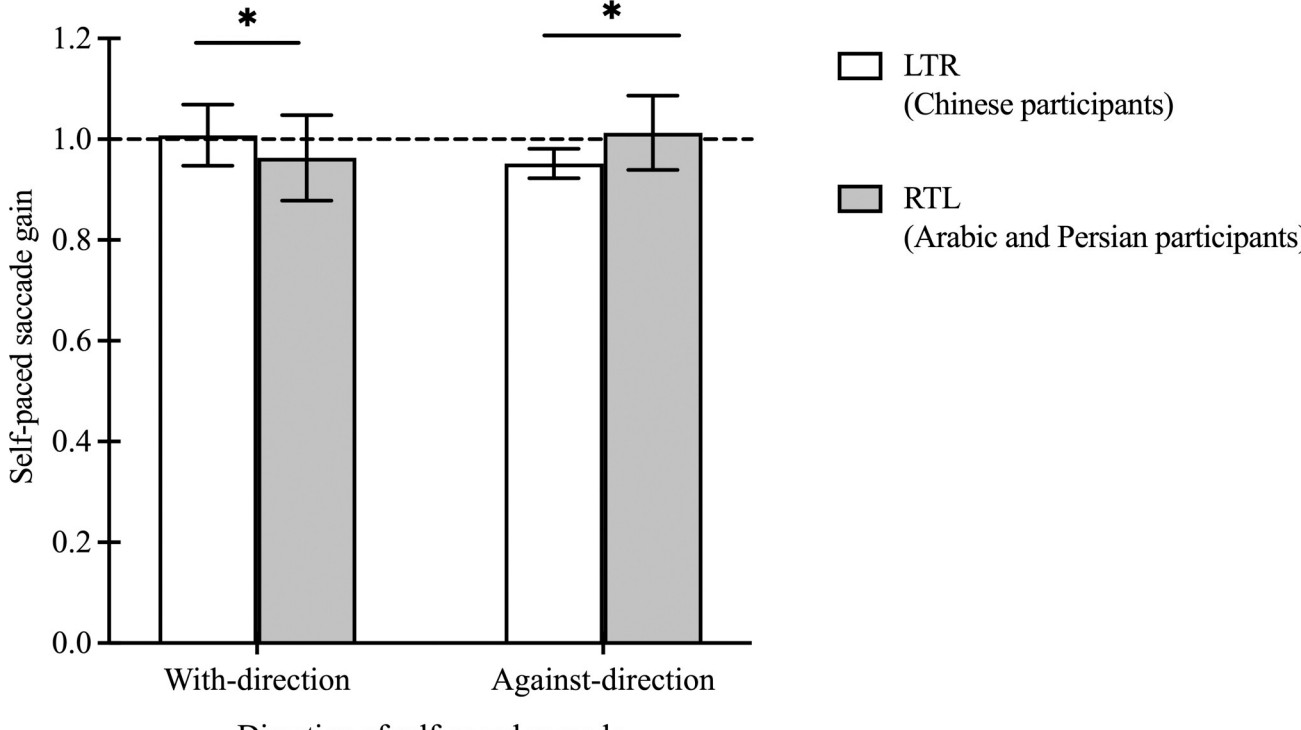

**Fig 6. Self-paced saccade gain towards 2 directions in LTR (Chinese participants) and RTL (Arabic and Persian participants) groups.** The self-paced saccade gain for the Chinese as well as the Arabic and Persian groups. Each bar represents the average self-paced saccade gain with standard deviations as the error bar. Black horizontal dotted line represents a gain equal to 1, where self-paced saccades land exactly on the target. Black horizontal line is the significant difference between the 2 saccade directions.

towards their non-habitual reading direction (i.e. towards the stimulus located at the right side of the monitor) (1.01 ± 0.07 vs. 0.96 ± 0.08, p = 0.04).

## 4. Discussion

The objective of this pilot study was to evaluate the impact of the primary habitual reading direction on the directionality of saccadic eye movements to low-cognitive-demand stimuli in young and healthy Chinese as well as Arabic and Persian participants using prosaccade, antisaccade and self-paced tasks. One of the major findings was the significantly shorter prosaccade latency of the Chinese participants whose primary habitual reading direction was from left to right (LTR) compared with that of the Arabic and Persian participants whose primary habitual reading direction was from right to left (RTL) when stimuli were presented along their habitual reading direction. However, the effect of reading direction on the antisaccade latency disappeared, where participants in both groups had similar latencies of accurate antisaccades. The second major finding was that the Chinese participants generated marginally but significantly more directional errors compared with the Arabic and Persian subjects when the stimuli appeared along their habitual reading direction in the antisaccade task.

### 4.1 Impact of habitual reading direction on prosaccade latency

In this study, we hypothesized that participants would produce shorter prosaccade latency to stimuli that appeared in their non-habitual reading direction (i.e. left for the Chinese participants and right for the Arabic and Persian participants). This was, however, only found in the

Arabic and Persian participants responding to the 5˚ stimuli. Previous studies reported that the direction of the stimuli presentation did not significantly affect the prosaccade latency of young participants, although these studies did not consider the participants' reading direction [33, 41]. In addition, our study found that Chinese readers had 16% shorter prosaccade latency than Arabic and Persian readers when the stimuli appeared 5˚ along their habitual reading direction. Amatya et al. reported that Chinese participants generated more low latency 'express saccades' compared with non-Chinese participants in an overlap prosaccade task [42]. It was argued that this difference in saccade latency should be attributed to human genetic diversity rather than cultural differences, as those Chinese participants who grew up in the UK also showed the same pattern of saccade latency as those who lived in mainland China [43]. A similar study by Knox and colleagues evaluated the antisaccade performance of Chinese participants and found a significantly higher antisaccade directional error rate in those Chinese participants who exhibited a higher proportion of express saccades [44]. They suggested that there was a difference in neurophysiological substrate concerned with eye movement that was not associated with culture. Nevertheless, in addition to express saccade latency, few studies have demonstrated the difference in normal reflexive saccades with both top-down and bottom-up control between populations. An electrophysiological study in primates showed that a neural signal took around 40 msec to be transmitted from the retina to the superior colliculus (SC), and it took approximately 20 msec to stimulate the SC to trigger a saccadic eye movement to a specific location [45]. However, the typical latency of a prosaccade is around 200 msec in humans [8]. Carpenter argued that such a long latency of the saccadic eye movement was due to the decision time on making a decision to look at the target or not [46]. One possible explanation for the different reaction times across groups was that these two groups' participants used different decision-making strategies, resulting in different decision-making times. However, further study is required to investigate the decision-making time for simple cognitive tasks between different populations.

It has been suggested that attention is needed to orient the target location prior to the execution of a saccade [12]. Pollatsek and colleagues measured the perceptual span in bilingual Israeli readers who spoke English as their second language. They found that bilingual Israeli readers showed an asymmetry perceptual span that extended 14 characters to the left of fixation and 4 characters to the right while reading Hebrew [47]. Additionally, although no overall extent of the perceptual span was examined, Jordan et al., reported a leftward asymmetry in perceptual span when participants read Arabic [48]. McConkie and Rayner reported that skilled readers of English and other alphabetic languages reading from left to right showed an asymmetric perceptual span, extending 14–15 characters positions to the right of fixation and 3–4 characteristics to the left [49]. In contrast, Chinese readers showed a narrower perceptual span that extended 1 character space leftward and 3 characters spaces rightward as reported in Inhoff and Liu [50] or extended beyond 4 characters spaces rightward depending on the font size as reported in Yan et al. [51]. It is possible that the early disengagement of attention in Chinese participants leads to a reduction in prosaccade latency because the 5˚ stimuli in the present study exceed the perceptual span used to acquire useful information for Chinese participants (the size of each character in [50] study was 0.9˚), but still fall into the perceptual span of the Arabic and Persian participants. Accordingly, no group difference was observed when the stimuli appeared against the participants' habitual reading direction, as the stimuli magnitude exceeded their perceptual span. Future study shall record the prosaccade latency using smaller magnitude stimuli (i.e. less than 5˚) or assess readers of other alphabetic languages to examine the effect of perceptual span on prosaccade latency.

## 4.2 Impact of habitual reading direction on prosaccade gain

Consistent with previous studies [36, 52, 53], we found a significant gain difference related to stimulus magnitude, in which participants undershot towards more distant stimuli. A significant interaction between group and stimulus direction was found in prosaccade gain. Arabic and Persian participants tended to make more accurate and larger prosaccades towards the side of their non-habitual reading direction (i.e. right side of the fixation), although this did not reach a significance level. This finding was consistent with the result as reported in [36] that rightward prosaccades had larger amplitude. Nevertheless, no primary habitual reading direction effect was found on prosaccade gain.

## 4.3 Impact of habitual reading direction on antisaccade latency

The current result agreed with previous findings that the latency of correct horizontal antisaccades was independent of stimulus direction or magnitude [36, 54, 55]. Although different eye movement paradigms (overlap and gap conditions) were tested on different participant cohorts, the latency of the correct antisaccades did not show the same pattern as the prosaccades. The study reported by Knox et al. also revealed that the correct antisaccade latency was identical between Chinese participants who exhibited a high proportion of express saccades and those who did not [44]. Reading relies more on perceptually driven saccades [56]. In contrast, cognition is needed to inhibit the reflexive error that was stimulated by a perceptual stimulus in the antisaccade task [8]. Therefore, it is possible that the cognitive difference induced by subjects' habitual reading habits had a greater influence on reflexive saccades compared with volitional saccades. That is, the reaction time of a reflexive saccade was more impacted by the reading direction than the initiation time of an antisaccade. Therefore, the correct antisaccade latency was not significantly affected by the habitual reading direction or the direction of the stimulus in the current study.

## 4.4 Impact of habitual reading direction on antisaccade error rate

The antisaccade error rate was found to be significantly higher in the Chinese participants compared with the Arabic and Persian participants for the with-direction stimuli presentation. The current study supported the parallel nature of saccade programming in the antisaccade task, which assumes a competition between the exogenous prosaccade generation and the endogenous antisaccade initiation at the onset of stimulus [5–7]. Therefore, the presence of a higher rate of directional errors in the Chinese group when the stimuli appeared at the side of their habitual reading direction might be attributed to the shorter latency of the prosaccade generation, as the faster prosaccades were more likely to reach the threshold first. Nevertheless, although Arabic and Persian participants had shorter prosaccade latency towards the 5˚ stimuli appearing at their non-habitual reading side, they did not show a higher antisaccade error rate, or more antisaccade errors compared with the 5˚ stimuli presenting at the right to the fixation. This implies that this quicker prosaccade latency would not result in making more antisaccade errors.

A previous study showed that prosaccade training increased the number of antisaccade errors because the reinforcement of the practice made it harder to inhibit a reflexive glance [57]. This may explain why more directional errors were made when stimuli were presented 5˚ to the fixation compared with that appeared 10˚ to the center. As reading involves smaller amplitude saccades along one direction and this practice might increase the chances of making antisaccade directional errors.

## 4.5 Impact of habitual reading direction on self-paced saccade

The self-paced saccade task has been considered as an almost entirely volitional eye movement task as no reflexive cues are presented to trigger saccadic eye movements [9]. The generation of self-paced saccades requires a series of quick volitional engagements and disengagements of attention between 2 static stimuli. Although it has been proposed that language processing drives the disengagement and shift of attention to the next word of interest in the direction of reading [58], the current result failed to find a difference in the mean inter-saccadic interval between the self-paced saccades initiated to the side of subjects' habitual reading direction and those to the non-habitual reading direction in both groups. One possible explanation is that the amplitude of the saccades required to execute the self-paced saccade task is much larger than those produced during normal reading, thus the difference was not shown in the current ocular motor task. Alternatively, it is possible that the subjects' sustained task engagement was more relevant to the performance of the self-paced saccade task, compared with the attentional modulation, as a result of the need to continuously initiate and execute eye movements [59]. Therefore, the inter-saccadic interval was not significantly different between groups. However, both groups' participants showed more accurate gain when they made a self-paced saccade towards the stimulus located on the right side. This result was similar with the prosaccades in those participants having larger amplitude saccades when the target appeared on the right side.

## 4.6 Limitations of the study

At present, very few studies have investigated the differences in saccadic eye movements in response to a low-cognitive demanding target between populations or individuals from different cultural backgrounds. The analysis of this pilot study has been primarily concentrated on the effect of habitual reading direction on the directionality of saccadic eye movements. However, only the Arabic and Persian subjects who made prosaccade to 5˚ stimuli supported our hypothesis, Nevertheless, we would like to point out several limitations in the current experiment. First, we did not recruit monolingual Arabic or Persian participants. The lack of monolingual subjects who only read from right to left leads to an uncertainty about the impact of habitual reading direction on saccadic eye movements as the Arabic and Persian participants in the present study were experienced in reading both directions. Second, we did not recruit participants in addition to Chinese who habitually read from left to right (alphabetic languages such as English). Therefore, it is difficult to determine if some differences in the current study were due to cultural, reading habit differences, different perceptual spans, or different orthographies, since Chinese is composed of strokes arranged within a square configuration, whereas phonetically based languages are written in a small number of letters with simple spatial forms. The difference in the perceptual span between the two languages has been discussed above (see Section 4.1). Finally, the current study had a relatively small sample size, limiting the generalizability of our result. However exploratory, this study offers some insights into the differences in saccade generation among different cultures. Further study can be performed on a large-scale cohort.

## 5. Conclusions

In the current pilot study, we aimed to investigate the effect of primary habitual reading direction on the directionality of the characteristics of saccadic eye movements in healthy Chinese as well as Arabic and Persian participants using prosaccade, antisaccade and self-paced tasks. We hypothesised that participants should show shorter prosaccade latency and a higher antisaccade error rate when stimuli were presented at the side of their non-habitual reading direction. Our hypotheses were partially accepted, with significantly shorter prosaccade latency

found in the Arabic and Persian participants in responding to the 5° rightward stimuli. The present study may contribute to the investigation of cultural or reading influences on the saccade generations between populations, and provide insights into the modulation of the correlation between saccades and attention.

## Supporting information

**S1 Appendix. Comparison of results between 2 eye trackers (Eyelink 1000 vs. Eyelink Portable Duo).**
(DOCX)

## Author Contributions

**Formal analysis:** Anqi Lyu, Larry Abel, Allen M. Y. Cheong.

**Methodology:** Anqi Lyu, Larry Abel, Allen M. Y. Cheong.

**Supervision:** Larry Abel, Allen M. Y. Cheong.

**Writing – original draft:** Anqi Lyu.

**Writing – review & editing:** Larry Abel, Allen M. Y. Cheong.

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
