## [Decision Letter · Decision Letter 0]

12 Jan 2023

PONE-D-22-23748Effect of habitual reading direction on saccadic eye movements: A pilot studyPLOS ONE

Dear Dr. Cheong,

Thank you for submitting your manuscript to PLOS ONE. After careful consideration, we feel that it has merit but does not fully meet PLOS ONE’s publication criteria as it currently stands. Therefore, we invite you to submit a revised version of the manuscript that addresses the points raised during the review process.

We look forward to receiving your revised manuscript.

Kind regards,

Shrikant R Bharadwaj

Academic Editor

PLOS ONE

and https://journals.plos.org/plosone/s/file?id=ba62/PLOSOne_formatting_sample_title_authors_affiliations.pdf.

Additional Editor Comments:

The manuscript has been reviewed by two experts in the field and their comments are appended below for the kind perusal of the authors. Both reviewers express interest in the study, but have reservations in terms the primary aim of the study, underlying hypothesis and the methodology. These need to be clarified in the revised version of the manuscript.

Reviewers' comments:

Reviewer's Responses to Questions

**Comments to the Author**

1. Is the manuscript technically sound, and do the data support the conclusions?

Reviewer #1: Partly

Reviewer #2: No

2. Has the statistical analysis been performed appropriately and rigorously? 

Reviewer #1: Yes

Reviewer #2: Yes

3. Have the authors made all data underlying the findings in their manuscript fully available?

Reviewer #1: Yes

Reviewer #2: Yes

4. Is the manuscript presented in an intelligible fashion and written in standard English?

Reviewer #1: Yes

Reviewer #2: Yes

5. Review Comments to the Author

Reviewer #1: In this manuscript, Lyu et al. investigate the difference in simple saccade tasks between 2 populations: Chinese readers and Arabic/Persian readers. Ostensibly, the comparison is between left-to-right and right-to-left reading.

The experiments appear to be sufficiently executed and the analyses appear sound. The writing is fine, with some suggestions below.

However there are a few possible confounds, some of which are mentioned in the "Limitations", but I think some points need to be reiterated.

For example, there are clear differences between the two languages utilized, beyond just the direction: the very different perceptual spans, and different orthographies. These are mentioned in different places in the text, but I think they should re-appear in Section 4.6 Limitations.

- When mentioning the Chinese participants (L458-459): "..differences could be due to the culture or reading habit differences", you could have: "culture, reading habit differences, different perceptual spans and different orthographies."

L463: "neural activities of oculomotor behaviours"

L474: "neural mechanisms of oculomotor behaviours"

-> I don't see how your study can tell us much about neural mechanisms (neural activities sounds strange), although it is insightful about oculomotor patterns/behaviours. I would prefer backing off the "neural" or just removing these lines.

SUGGESTIONS

L145: "(also see S1 Appendix)." You could explicitly state in the text something like "there was no statistical difference between the results from the two eye trackers (see S1 Appendix)", rather than having the reference to the Appendix without any clue about what is provided there.

L148: I assume the tasks must have been blocked, but it would be good to state so explicitly.

L340: "Chinese participants generated more low latency compared to non-Chinese participants (Caucasian participants") -> does 'Caucasian participants' add any useful information here? I would remove.

MINOR/GRAMMAR

Abstract, 3rd-to-last-sentence: "for stimulus appearing" -> "for stimuli appearing"

L55: "see below of section 1.3" -> see section 1.3 below

L90: "Yan et al. took the advantage that" -> "Yan et al. took advantage of the fact that"

L108: "..compared with the target appeared.." -> "compared with when the target appeared"

L156: "whose latency falls between" -> "whose latency fell between"

L209: "The prosaccade latency for the LTR and RTL groups responded to 5o"

L235: (same error as above)

L248: (same error as above)

L267: (same error as above)

Reviewer #2: Cheong et al., have investigated the habitual reading direction and its effect on saccadic eye movements. They are presenting their pilot study results here. While the authors give a nice introduction on saccades, they are unable to articulate what exactly is their study rationale. There is only one line that states this study result will help to better investigate the differences of eye movement control across populations! Why is this important? If you are projecting at a population level, you need to justify why that is the case, or are you asking a basic science question to explain some mechanism. So I am not clear at all with the study rationale here.

I am also confused on the study hypothesis. The abstract states saccades in the direction of habitual reading direction would show longer prosaccade latency. Shouldn’t this be shorter? I suppose with reading one could argue both ways, as one begins to read from right to left, more saccades are planned leftwards, but the reverse sweep saccade needs to land back to right. Under this premise it would have been interesting to check the gain of the anti-saccades & check for this direction bias. I suppose one could then hypothesize that the antiscaccade gain in the direction of the reverse sweep saccade perhaps is more accurate.

I am not sure on the study design, and why those amplitudes were chosen. The primary question is hinging on the reading direction and influence of that on saccades. In that case, the participants chosen should have been tested for their reading

pattern on a reading task to check for reading speed, regression saccades, reading comprehension, and perhaps even eye movement recordings to make sure they are standardized, especially when the sample size is small. The study design does not explain why the amplitudes of 5 and 10 degrees were chosen. Why are we looking at self-paced saccades etc.

Overall, I am not fully convinced with this study question and the design used to answer that question. If it were to be a pilot study, still the design should be robust. I am lost there.

* I give a disclaimer that I am reviewing this paper for the first time. I understand it has been reviewed earlier and some comments of the earlier reviewer have been addressed in this version that I am reading. I realize more figures have been added as a result, but I feel that only figures that are useful to generate new knowledge or are surprising can be retained. Not every result needs a figure!

6. PLOS authors have the option to publish the peer review history of their article (what does this mean?). If published, this will include your full peer review and any attached files.

Reviewer #1: No

Reviewer #2: No

---

## [Author Response · Author response to Decision Letter 0]

10 Feb 2023

Please refer to the attached document for our respond to reviewers' comments.

---

## [Decision Letter · Decision Letter 1]

28 Mar 2023

PONE-D-22-23748R1Effect of habitual reading direction on saccadic eye movements: A pilot studyPLOS ONE

Dear Dr. Cheong,

Thank you for submitting your manuscript to PLOS ONE. After careful consideration, we feel that it has merit but does not fully meet PLOS ONE’s publication criteria as it currently stands. Therefore, we invite you to submit a revised version of the manuscript that addresses the points raised during the review process.

We look forward to receiving your revised manuscript.

Kind regards,

Shrikant R. Bharadwaj

Academic Editor

PLOS ONE

Journal Requirements:

Additional Editor Comments:

The authors have addressed all the comments of the reviewers. There is only a minor issue concerning accessibility to the raw data. While the raw data is accessible through the link provided by the authors, could the authors also share the any software that may be written to analyze the data and generate the figures in the manuscript. This will be very useful for anyone else attempting to replicate the data or re-analyze the results.

Reviewers' comments:

Reviewer's Responses to Questions

**Comments to the Author**

1. If the authors have adequately addressed your comments raised in a previous round of review and you feel that this manuscript is now acceptable for publication, you may indicate that here to bypass the “Comments to the Author” section, enter your conflict of interest statement in the “Confidential to Editor” section, and submit your "Accept" recommendation.

Reviewer #2: All comments have been addressed

2. Is the manuscript technically sound, and do the data support the conclusions?

Reviewer #2: Yes

3. Has the statistical analysis been performed appropriately and rigorously? 

Reviewer #2: Yes

4. Have the authors made all data underlying the findings in their manuscript fully available?

Reviewer #2: Yes

5. Is the manuscript presented in an intelligible fashion and written in standard English?

Reviewer #2: Yes

6. Review Comments to the Author

Reviewer #2: My concerns have been addressed by the authors. There are minor errors, which I think in proof reading can be fixed.

7. PLOS authors have the option to publish the peer review history of their article (what does this mean?). If published, this will include your full peer review and any attached files.

Reviewer #2: No

---

## [Author Response · Author response to Decision Letter 1]

31 Mar 2023

Please refer to the attached document for our response.

---

## [Decision Letter · Decision Letter 2]

24 May 2023

Effect of habitual reading direction on saccadic eye movements: A pilot study

PONE-D-22-23748R2

Dear Dr. Cheong,

We’re pleased to inform you that your manuscript has been judged scientifically suitable for publication and will be formally accepted for publication once it meets all outstanding technical requirements.

Kind regards,

Nick Fogt

Academic Editor

PLOS ONE

Additional Editor Comments (optional):

Reviewers' comments:

Reviewer's Responses to Questions

**Comments to the Author**

1. If the authors have adequately addressed your comments raised in a previous round of review and you feel that this manuscript is now acceptable for publication, you may indicate that here to bypass the “Comments to the Author” section, enter your conflict of interest statement in the “Confidential to Editor” section, and submit your "Accept" recommendation.

Reviewer #2: All comments have been addressed

2. Is the manuscript technically sound, and do the data support the conclusions?

Reviewer #2: Yes

3. Has the statistical analysis been performed appropriately and rigorously? 

Reviewer #2: Yes

4. Have the authors made all data underlying the findings in their manuscript fully available?

Reviewer #2: Yes

5. Is the manuscript presented in an intelligible fashion and written in standard English?

Reviewer #2: Yes

6. Review Comments to the Author

Reviewer #2: There were only minimal revisions required in this article. The authors have addressed it. They have provided updated references and have shared their data sheet for replication. I don't have other concerns with this manuscript.

7. PLOS authors have the option to publish the peer review history of their article (what does this mean?). If published, this will include your full peer review and any attached files.

Reviewer #2: No

---

## [Editor Report · Acceptance letter]

26 May 2023

PONE-D-22-23748R2 

Effect of habitual reading direction on saccadic eye movements: A pilot study 

Dear Dr. Cheong:

I'm pleased to inform you that your manuscript has been deemed suitable for publication in PLOS ONE. Congratulations! Your manuscript is now with our production department. 

Kind regards, 

on behalf of

Dr. Nick Fogt 

Academic Editor

PLOS ONE